# Production of Xylooligosaccharides from *Jiuzao* by Autohydrolysis Coupled with Enzymatic Hydrolysis Using a Thermostable Xylanase

**DOI:** 10.3390/foods11172663

**Published:** 2022-09-01

**Authors:** Liqin Qin, Jinghao Ma, Huafeng Tian, Yanli Ma, Qiuhua Wu, Shuang Cheng, Guangsen Fan

**Affiliations:** 1School of Food and Health, Beijing Technology and Business University (BTBU), Beijing 100048, China; 2College of Chemistry and Materials Engineering, Beijing Technology & Business University (BTBU), Beijing 100048, China; 3Henan Key Laboratory of Industrial Microbial Resources and Fermentation Technology, Nanyang Institute of Technology, Nanyang 473004, China; 4Beijing Engineering and Technology Research Center of Food Additives, Beijing Technology & Business University (BTBU), Beijing 100048, China

**Keywords:** *Jiuzao*, autohydrolysis, thermostable xylanase, xylooligosaccharides, enzymatic hydrolysis

## Abstract

The production of xylooligosaccharides (XOS) from *Jiuzao* was studied using a two-stage process based on autohydrolysis pretreatment followed by enzymatic hydrolysis. *Jiuzao* was autohydrolyzed under conditions where temperature, time, particle size, and solid-liquid ratio were varied experimentally. Optimal XOS production was obtained from *Jiuzao* with a >20 mesh particle size treated at 181.5 °C for 20 min with a 1:13.6 solid-liquid ratio. Subsequently, optimal enzymatic hydrolysis conditions for xylanase XynAR were identified as 60 °C, pH 5, and xylanase XynAR loading of 15 U/mL. Using these conditions, a yield of 34.2% XOS was obtained from *Jiuzao* within 2 h. The process developed in the present study could enable effective and ecofriendly industrial production of XOS from *Jiuzao*.

## 1. Introduction

As one of the world’s most famous distilled spirits, *Baijiu* is known for its unique flavor [1]. China is the major producer of *Baijiu* and its production amounted to 7.4 million kiloliters in 2020 [2]. *Baijiu* is made with solid-state fermentation and distillation using sorghum, corn, other grains, and rice hulls [3,4]. A mixture of grain and rice hull residues, known as *Jiuzao*, is left after the solid-state fermentation and solid-state distillation processes are complete. Because eight units of *Jiuzao* are generated during the production of one unit of *Baijiu*, nearly 100 million tons of solid waste are generated each year from *Baijiu* production, creating an environmental concern since *Jiuzao* decay produces a foul odor [2]. Because the main components of dry *Jiuzao* are cellulose, hemicellulose, and lignin, it represents a valuable resource for the production of value-added products such as xylooligosaccharides (XOS). The use of *Jiuzao* to produce XOS would be economically and environmentally beneficial.

Currently, XOS derived from xylan-containing lignocellulosic biomass is widely used in food, chemical, pharmaceutical, feed, and nutraceutical industries [5,6,7]. Although, many different approaches have been reported for XOS production from various sources of biomass, the most effective method employs xylanase to hydrolyze xylan, the main component of hemicellulose [7,8]. This method is not only environmentally friendly, but also produces the main functional components, xylobiose (X2) and xylotriose (X3) [7]. Thus, the enzymatic degradation of hemicellulose in *Jiuzao* to produce XOS should be possible. However, any such new production process must overcome two problems that limit the accessibility of components to hydrolytic enzymes: the strong crystalline structure of the cellulose, and the complex structure crosslinking lignin and hemicellulose with cellulose in *Jiuzao* [7,9]. Effective pretreatments are needed to disrupt the crystalline structure of cellulose and increase hemicellulose exposure to hydrolytic enzymes thereby increasing the efficiency of enzymatic hydrolysis and XOS yield [10,11]. Among many pretreatment methods, autohydrolysis or hot-compressed water pretreatment uses water as the only reagent and is preferred for its economic and environmental benefits [11,12,13,14].

Autohydrolysis pretreatment, in which biomass is exposed to pressurized hot water (100–240 °C) extract hemicelluloses into the water phase, is one of the most promising pretreatment methods [15,16,17]. Water under pressure penetrates the cell walls of the plant biomass, hydrates cellulose, and dissolves hemicellulose and lignin. In addition, the lignocellulosic structure is disrupted by the acidity of the water at high temperature (around 200 °C) and the organic acids released from the hemicellulose [12,18,19]. Autohydrolysis does not require the addition of any chemical, generates reactive cellulose fiber, enables recovery of most of the pentosans, and generates degradation products that do not significantly inhibit subsequent hydrolysis and fermentation [12,16,20]. The fundamental characteristics of the autohydrolysis of lignocellulosic biomass for sugar production have been described in general [20,21]. However, specific features of the process need further clarification to optimize this promising technology, including the details related to both sugar production and inhibitor formation during the hot-water extraction process. Different biomass types have different structure and composition, which would likely require different pretreatment conditions. Thus, the pretreatment process must be tailored to the unique compositional and structural features of the lignocellulosic biomass being used [11].

For production of XOS in the present study, *Jiuzao* was pretreated with autohydrolysis and then hydrolyzed with XynAR, a thermostable xylanase obtained by molecular modification in our laboratory. A method of production of XOS from *Jiuzao* by autohydrolysis with xylanase was established, which can provide a scheme for generating high value products from *Jiuzao*.

## 2. Materials and Methods

### 2.1. Materials

*Jiuzao* was provided in 2019 by the Heibei Bancheng Liquor Group, which is an enterprise producing strong-flavor *Baijiu*, dried at 60 °C for 3 d, and stored in plastic containers at room temperature. Beechwood xylan from Sigma Chemical Company (St. Louis, MO, USA) was used. XOS standards, including X2 (≥95%, HPLC), X3 (≥95%, HPLC), xylotetraose (X4) (≥95%, HPLC), and xylopentaose (X5) (≥95%, HPLC) were obtained from Megazyme (Ireland). All other chemicals and laboratory reagents, unless stated otherwise, were reagent grade and obtained from commercial sources.

### 2.2. Jiuzao Composition

*Jiuzao* moisture content was calculated using the weight before and after drying at 105 °C. Ash content was measured by igniting in a muffle furnace according to AOAC 923.03. Starch content was determined with the α-amylase method as described in AOAC 996.11. *Jiuzao* that had been completely hydrolyzed with α-amylase to remove starch as described previously [22] was used to assess carbohydrate and lignin content with the method of the National Renewable Energy Laboratory (NREL) [23]. Briefly, 300 mg of *Jiuzao* that had been pretreated with α-amylase was hydrolyzed with 3 mL of 72% (*w*/*w*) sulfuric acid for 1 h at 30 °C, diluted to 4% (*w*/*w*) acid solution by adding 84 mL of double distilled water, and autoclaved at 121 °C for 1 h. The autoclaved samples were filtered through 0.45 µm membranes to separate solid and liquid phases. After thoroughly rinsing with water, the residual biomass was dried at 60 °C to a constant weight. The total lignin content was estimated as acid-soluble and insoluble lignin. The acid-soluble lignin was determined from absorbance at 205 nm with a UV-Vis spectrophotometer (TU-1901, Pgeneral, UV Spectrophotometer). The insoluble lignin in the residual dry solid biomass was estimated gravimetrically by subtracting the residual ash weight obtained after heating at 575 °C for 4 h in a muffle furnace from the total dry weight obtained by drying the original sample at 105 °C to constant weight. Monomeric sugar content in the liquid hydrolysate was used to quantify cellulose and hemicellulose. After neutralizing with CaCO_3_ and filtering through a 0.22 µm membrane, glucose, xylose (X1), and arabinose (A) in the hydrolysate were analyzed by high-performance liquid chromatography (HPLC, Shimadzu. Co., Ltd., LC-10AD, Kyoto, Japan) with a refractive index detector (RID).

### 2.3. Autohydrolysis Processing

The autohydrolysis reactor was a stainless-steel autoclave with a working volume of 600 mL (BR-300, JULABO, Seelbach, Germany) and a temperature controller. Thirty-six grams of *Jiuzao* were mixed with 360 mL deionized water and heated with an electric mantle heater while being continuously stirred at 300 rpm with a paddle agitator. When a target temperature was reached, the pretreatment temperature was maintained for 20 min at 180 °C with a PID controller. Upon completion of the set time, the reactor was cooled with cold water. After pretreatment, the solid fraction (SF) and the liquid fraction (LF) were separated by filtration and the solid residue washed several times with deionized water. The SF was then lyophilized, and the LF including wash water was stored at −20 °C for enzymatic hydrolysis. The weight of the SF and the final volume of the LF were measured. 

The parameters of temperature, time, *Jiuzao* particle size, and solid-liquid ratio were optimized first by varying single factors as follows: temperatures of 160, 170, 180, 190, and 200 °C, pretreatment times of 0, 20, 40, 60, and 80 min, particle derived from sieving through meshes of >20, 20–40, 40–65, and 65–80, and the solid-liquid ratios of 1:8, 1:10, 1:12, 1:14 and 1:16. According to the single factor experimental results, three factors including temperature, time, and solid-liquid ratio were selected for response surface experiments using the Box–Behnken experimental design (BBD, Design-Expert Software 11.0, StatEase Inc., Minneapolis, MN, USA). The response value was XOS yield after enzymatic hydrolysis of the sample pretreatment by autohydrolysis. 

To evaluate the combined effect of temperature (T) and time (t) on thermal treatment efficiency, logarithmic values of the severity factor (log R_o_) were calculated for autohydrolysis treatments as expressed in Equation (1) [24].
(1)$$R_{0}=\int_{0}^{t}\exp\frac{T-T_{\mathrm{ref}}}{w}\mathrm{dt}$$

Here, t is the holding time (min) at the treatment temperature T, and T_ref_ is the reference temperature where little or no reaction occurs in the system being studied. A fitted value of 14.75, which is an empirical value related to activation energy of the reaction, is used for the arbitrary constant w. A value of 100 °C was used as the reference temperature. 

### 2.4. XynAR Production and Enzyme Assay

Xylanase XynAR, modified from XynA was expressed in *Escherichia coli* (*E. coli*). XynAR production and activity were assessed as described previously [25]. Briefly, the transformants were cultured at 37 °C for 12 h in Luria-broth medium including 40 μg/mL kanamycin. Isopropyl β- D-1-thiogalactopyranoside at a final concentration of 0.5 mmol/L was added into the culture to induce XynAR expression. *E. coli* cells were obtained by centrifugation and disrupted by ultrasonication to get crude XynAR. The activity of XynAR was assayed at 60 °C with 1.0% (*w*/*v*) beechwood xylan (50 mmol/L citrate buffer, pH 6.0) for 10 min as in previous study [25]. The amount of reducing sugar liberated was determined by the 3,5-dinitrosalicylic acid method using xylose as the standard. One unit (U) of enzyme activity was defined as the amount of enzyme required to produce 1 μmoL of reducing sugar per minute.

### 2.5. Enzymatic Hydrolysis 

Preliminary experiments showed that a mixture of the SF and LF yielded more XOS after enzymatic treatment than did the SF alone. Therefore, to obtain maximum XOS yield and improve experimental repeatability, both the SF and LF were used in subsequent work. The enzymatic hydrolysis with XynAR was performed as described previously [26]. In detail, enzymatic hydrolysis was performed on 15 mL samples. The amount of SF required for each 15 mL of LF was calculated to match the proportion in the original autohydrolysis product. After mixing each portion of SF with 15 mL of LF (SF-LF), the 15 mL mixtures were used for enzymatic hydrolysis reactions at 60 °C. After adjusting pH to 6.0, XynAR with a final concentration of 5 U/mL was added to the LF-SF mixture for 2 h. After hydrolysis, the samples were boiled at 100 °C to inactivate the enzyme. The formation of XOS was monitored using HPLC. The influence of different conditions such as temperature (55, 60, 65, 70, and 75 °C), pH (3, 4, 5, 6, and 7), enzyme concentration (5, 15, 45, 135, and 405 U/mL), and hydrolysis period (30, 60, 120, 240, and 480 min) on XOS yield was assessed in single factor experiments. For further optimization a four factor, three-level orthogonal experiment (L_9_(3^4^)) was done using Design-Expert Software 11.0. Based on the results of the single factor tests, nine enzymatic experiments were carried out at temperatures of 60, 65, and 70 °C, pH 3, 4, and 5, enzyme concentrations of 5, 15, and 25 U/mL, and hydrolysis periods of 2, 4, and 6 h. 

### 2.6. Analytical Methods

The concentration of XOS was analyzed using a LC-10AD HPLC (Shimadzu, Japan) equipped with a Shodex Sugar KS-802 packed column (8 mm ID ×300 mm, F6378020) and a RID-10A according to Wu et al. [26]. The column was maintained at 65 °C and eluted with deionized water at a flow rate of 0.8 mL/min. The concentration of glucose, xylose, and arabinose was determined by HPLC system equipped with Aminex HPX-87H column (Bio-Rad, Hercules, USA). The column temperature was kept at 65 °C and the mobile phase was 5 mmol/L H_2_SO_4_ at a flow rate of 0.6 mL/min.
P(X) (%) = C(X) (g/L) × V (L) × 100%/W (g)(2)
where P(X) is the XOS, xylose, glucose, or arabinose yield, C(X) is the XOS (DP = 2–5), xylose, glucose, or arabinose concentration in the reaction mixture, V is the volume of LF (L), and W is the hemicellulose or total of cellulose and starch weight in the test sample (g).

### 2.7. Fourier Transform Infrared Spectroscopy (FTIR)

FTIR experiments (Nicolet™ iS™50, Thermo Scientific™, Waltham, MA, USA) were done using a Fourier transform infrared spectrophotometer equipped with a detector (DTGS) operating at 4000–400 cm^−1^ spectral range at resolution 4 cm^−1^ and 32 scans per sample. About 5 mg of finely milled raw materials and solid residues were used for FTIR analysis.

### 2.8. Scanning Electron Microscopy (SEM)

Micrographs of samples were observed by scanning electron microscopy SEM (EVO 18, ZEISS, Jena, Germany) at 10 kV. Before observation, the samples were fixed to a 70 mm diameter specimen platform with double-sided adhesive carbon tape and coated with sputter palladium. Each micrograph was taken at different magnification.

### 2.9. Data Analysis

Each treatment was conducted in triplicate and the results were expressed as the mean. All statistical analyses were performed with Design-Expert Software 11.0 and Excel 2019 (Microsoft Corporattion, Washington, DC, USA).

## 3. Results and Discussion

### 3.1. Composition of Jiuzao

The primary components of *Jiuzao* were cellulose at 25.14% followed by lignin at 18.72%. The content of hemicellulose, a polysaccharide composed of xylan, was of special interest because XOS are obtained by its autohydrolysis and enzymatic hydrolysis. *Jiuzao* contained 13.08% hemicellulose, which is within the range of values previously reported [27,28]. In addition, *Jiuzao* also contained 10.08% starch, 11.54% ash, and 11.18% of other components. The various content values reported in the literature may be due to differences in feedstocks, *Baijiu* production process, and/or methods of measurement. Although the hemicellulose content of *Jiuzao* at the present study was lower than that reported for most other biomass sources, the yearly output of *Jiuzao* is large and it is the main pollutant generated during *Baijiu* production [29,30,31]. Using various *Jiuzao* components to produce value-added products will help mitigate this environmental problem. Hemicellulose would likely be the first component to be used commercially, with the gradual utilization of other components established through a step-by-step approach. The hemicellulose components in *Jiuzao* from various *Baijiu* production facilities could differ and thus the *Jiuzao* from some plants might be better for XOS production [27,28]. 

### 3.2. Optimum Conditions for Autohydrolysis

#### 3.2.1. Effect of Autohydrolysis Temperature on the Production of XOS and Monosaccharides

Previous reports showed that the yield of XOS depends on the autohydrolysis conditions, especially temperature and reaction time [12,17]. In the present study, the yield of XOS increased first and then decreased with increasing temperature during autohydrolysis and the highest yield was obtained at 190 °C (Figure 1). During the process of autohydrolysis, some treatment conditions produced an oligosaccharide, X2*, with a degree of polymerization (DP) between that of X2 and X3. X2* was obtained with HPLC according to the XOS analytical method. After treatment with 4% (*w*/*w*) sulfuric acid at 121 °C for 1 h, the monosaccharide composition of X2* determined by HPLC to be xylose and arabinose, which confirmed that it was an XOS containing arabinose. The production of this oligosaccharide was measured according to the standard curve for X2, and it was likely due to the complexity and structure of the fibrous materials and random interruption of the fibrous structure during autohydrolysis. Therefore, because of the distribution of arabinose residues in *Jiuzao* and their sensitivity to high temperature, XOS containing arabinose can only be produced under appropriate autohydrolysis conditions. In futures studies the preparation conditions, purification, the effects of X2* on probiotics and antioxidant function will be studied. A similar trend for xylose yield with XOS yield was reported previously [32]. As reported in other studies [12,33], with increasing autohydrolysis treatment temperature the hemicellulose components degrade into soluble oligosaccharides and monosaccharides, but as the temperature increases further the oligosaccharides and monosaccharides are completely degraded. In contrast, arabinose yield decreased with increasing temperature. Compared to xylose, arabinose degrades more easily because of the higher susceptibility of arabinosyl linkages to hydrolytic reactions [14,17,32,34]. An increased yield of arabinose was observed at 200 °C, possibly due to the presence of some hemicellulose components that are difficult to degrade as previously reported [14]. The change in glucose yield was complex, showing a decreasing trend followed by an increase and then another decrease. This result may be due to the complex composition of *Jiuzao*. Besides the components described previously, it contains some residual sorghum starch that was not used up during *Baiju* production. This starch is easily degraded by autohydrolysis, producing a relatively high yield of glucose at lower temperature (i.e., 160 °C). The glucose content then decreased as the temperature increased because it was degraded into by-products such as furfural. Finally, at higher temperatures the autohydrolysis of cellulose generated another increase followed by a decrease in glucose content. The proportion of XOS and xylose increased with increasing temperature then decreased, indicating that the degradation of hemicellulose by autohydrolysis may be a disordered process [35]. When the temperature is low, large soluble polymers and xylose are mostly produced as content of XOS with a DP of 2–5 is lower. With increasing temperature, the XOS production rate is higher than that of xylose, thus increasing the XOS-to-xylose ratio. However, with a further increase in temperature, the XOS degradation rate exceeds the production rate, and the degradation rate of xylose approximates the production rate, resulting in a decrease of the XOS-to-xylose ratio.

After enzymatic hydrolysis, the changed behavior of XOS yield is like that of the autohydrolysis treatment. The autohydrolysis treatment condition with the highest XOS yield changed from 190 °C to 180 °C compared to that without enzymatic hydrolysis, indicating the role of xylanase in XOS production. Enzymatic hydrolysis with xylanase can not only further improve the yield of XOS, but also reduce the autohydrolysis intensity compared to the autohydrolysis-only treatment. In addition, the yield of XOS produced by xylanase hydrolysis was higher at lower autohydrolysis treatment temperatures and with increasing autohydrolysis temperatures the yield of XOS produced by enzymatic hydrolysis showed a downward trend. According to previous reports, this result may be due to an increase in the type and concentration of by-products inhibiting xylanase activity [33,36,37,38]. This explanation does not rule out the possibility of product inhibition of a large amount of XOS produced by autohydrolysis. Because the xylanase XynAR used in the present study has good XOS production capacity and little activity in xylose production, the xylose yield showed little change compared to that without enzymatic hydrolysis and the changed behaviors consistent with the result without enzymatic hydrolysis. Thus, after enzymatic hydrolysis, the XOS to xylose ratio increased in various treatments compared to no enzymatic hydrolysis, indicating the role of xylanase in the production of XOS. Furthermore, based on the different changes in XOS and xylose yields after enzymatic hydrolysis, the ratio of XOS-to-xylose after enzymatic hydrolysis shows a changed trend unlike that with autohydrolysis treatment, and the treatment temperature producing the highest XOS-to-xylose ratio differed. The temperature for the higher XOS-to-xylose ratio was 170 °C coupled with enzymatic hydrolysis, lower than that of the autohydrolysis only treatment temperature of 180 °C. Considering the yield and relative purity of XOS, 180 °C was chosen for the autohydrolysis treatment.

#### 3.2.2. Effect of Autohydrolysis Time on the Production of XOS and Monosaccharides

The autohydrolysis hold time is another key factor affecting the degradation of fibrous materials [39]. Therefore, the effect of different holding times on XOS production from *Jiuzao* was analyzed at the optimal temperature of 180 °C. The change in XOS yield with holding time approximated that of the treatment temperature in autohydrolysis process (Figure 2). With the extension of holding time, XOS yield gradually increased and reached a high of 12.3% at the 60 min hold time, then trended downward over the next 60 min. In previous reports, the autohydrolysis hold time for the highest yield of XOS at 180 °C was less than that in the present study. For example, for sugarcane bagasse the time was 40 min, for hazelnut (*Corylus avellana* L.) shell 30 min, and for *Miscanthus* × *giganteus* 20 min [12,33,34]. The need for a greater hold time for *Jiuzao* may be due to partial degradation of hemicellulose during *Baijiu* production, resulting in more energy needed to destroy the relatively rigid parts of the molecule. Alternatively, other factors could be involved, such as solid-liquid ratio or particle size [14,35]. In the present study, the yield of xylose increased with increasing hold time, whereas the yield of arabinose decreased. This result is likely related to the composition characteristics of xylose and arabinose units in the fibrous parent materials. The glucose yield change showed an initial slight decrease followed by an increase. The mechanism of this change is probably the same as of the change produced by the different temperatures (Figure 1). Regarding the relative yields of XOS and xylose, the ratio of XOS-to-xylose increased first and then decreased. When the hold time was 60 min at 180 °C, the ratio peaked at 176.3%.

Although the changed behavior of the XOS yield with and without enzymolysis was similar, increasing at first and then decreasing with greater hold time, the yield of XOS after xylanase hydrolysis was higher than that without enzymatic hydrolysis. Moreover, the hold time for the highest yield of XOS was shorter in the autohydrolysis plus enzymatic hydrolysis treatment (at a hold time of 40 min) than that in the autohydrolysis treatment (at a hold time of 60 min). This relative result is like that obtained for the temperature optimization experiment in that the autohydrolysis plus enzymatic hydrolysis treatment peaked at a lower temperature. These results confirm the benefit of enzymatic hydrolysis for XOS preparation from fibrous materials, i.e., lower production cost from both improved yield and reduced autohydrolysis intensity. The peak and decline of XOS yield with different autohydrolysis treatment times after enzymatic hydrolysis parallels that for the different temperature conditions (Figure 2), and the possible mechanisms are the same. In addition, xylose content increased only slightly confirming the specific effect of enzyme hydrolysis. After enzymatic hydrolysis, the ratio of XOS-to-xylose at various hold times all increased compared to treatments lacking enzymatic hydrolysis, and the trend peaked then declined as the XOS and xylose yields changed. The ratio was the highest when the treatment holding time was 20 min, followed by 40 min. Considering both the yield of XOS and its ratio to xylose, the optimum hold time was 40 min. 

#### 3.2.3. Effect of Particle Size on the Production of XOS and Monosaccharides

During autohydrolysis, materials with differing particle size having slight differences in structural compactness and/or surface to volume ratio undergo different degrees of action by electrolyzed hydrogen ions and other substances, such as hydronium ions (H_3_O^+^) and acetic acid [39]. These differences affect the yield of XOS. In the present study, XOS yield after autohydrolysis fluctuated as particle size decreased (Figure 3), likely due to the complex fibrous structure in *Jiuzao*. Fibrous materials may contain xylan components that are easily degraded by autohydrolysis [14], producing XOS and other components easily even from large particles. As particle size decreases, a greater proportion of easily degraded components are available for conversion into XOS due to increased surface to volume ratio compared to the material with larger particle size (such as particle size >20 mesh). Thus, there would be sufficient energy available for the easily degradable parts to be converted in the material with smaller particle size (such as particle size 20–40 mesh), but the excess energy after degrading the easily degradable part is insufficient for production of new XOS by further degradation of the denser parts of the fibrous material and the complex and refractory xylan components. Nevertheless, the excess energy is sufficient to degrade the XOS produced by the easily degradable xylan components to produce other by-products. As a result, the yield of XOS was less when the particle size was 20–40 mesh compared to >20 mesh. As the particle size decreases further, partial components that are not easily degraded will also be destroyed by autohydrolysis to produce XOS thus improving the yield. With further reductions in particle size, the cycle will repeat. This process results in a layer-by-layer degradation of xylan in material as the particle size decreases, which may be due to the presence of relatively complex *Jiuzao* components, including a small amount of wheat and sorghum bran and a large amount of rice hulls, and the differences in fiber structure of these components [40]. In addition, the particle size for high XOS yield from *Jiuzao* treated by autohydrolysis was larger than that for materials used in other studies [33,41,42]. Perhaps the structure of *Jiuzao* was damaged to some extent because of microbial pretreatment and repeated cycles of heating during *Baijiu* production, rendering it more susceptible to autohydrolysis [43]. Xylose yield first decreased and then increased. The mechanisms producing variations in XOS yield across particle sizes are likely those for variations in xylose yield with particle size. The relative yield differences between the two are likely due to differences in the degree of degradation from xylan to XOS versus the degradation from xylan or XOS to xylose, resulting in a slight difference in the changed behavior of xylose yield. The ratio of XOS-to-xylose decreases with decreasing *Jiuzao* particle size, indicating the level of net increment of XOS increase (the total amount of xylan to XOS minus the amount of XOS converted to xylose or other by-products) is less than that of xylose increase (the total amount of xylan or XOS converted to xylose minus the amount of xylose converted to other by-products). With decreasing *Jiuzao* particle size, the changed trend of arabinose yield is like that of XOS yield, which may be related to its ease of degradation and the possible reasons for the XOS yield change trend [39]. The yield of glucose increased first and then decreased with decreasing particle size, which may be the result of relatively slow degradation of cellulose and some residual starch.

After xylanase hydrolysis, the yield of XOS increased to varying degrees for different particle sizes samples. This increase weakened as *Jiuzao* particle size decreased, perhaps due to the inhibition of xylanase hydrolysis by many by-products produced with small particle size material which supports our hypothesis regarding particle size effects for XOS yield. The ratio of XOS-to-xylose in the product after enzymatic hydrolysis was higher than in the absence of enzymatic hydrolysis. The ratio also decreased with the decrease of *Jiuzao* particle size, where greater by-products production caused a decrease in enzyme hydrolysis. Considering the yield and cost of XOS, particles of *Jiuzao* from the 20-mesh screen were selected as the optimum size for further experiments.

#### 3.2.4. Effect of Solid-Liquid Ratio on the Production of XOS and Monosaccharides

The solid-liquid ratio determines the concentration of hydrogen ions, hydronium ions, and acetic acid produced in autohydrolysis required to disrupt the fibrous structure of *Jiuzao*, which in turn affects the yield of XOS [39]. As the solid-liquid ratio increased, the yield of XOS fluctuated (Figure 4). The changed trend of xylose yield paralleled that of XOS, increasing initially, then decreasing, and then increasing again. Although the changed trends of XOS and xylose were the same, the ratio of both shows the opposite trend. This result may be due to different effects of hydrogen ions, hydronium ions, and acetic acid on xylan, XOS, and xylose, resulting in different degrees of net increase for XOS and xylose. The change of glucose yield was like that of XOS, but the inflection point was different due to the different composition and structure of starch and cellulose and the different sensitivity to hydrogen ions, hydronium ions, and/or acetic acid. In addition, the yield of arabinose shows the opposite trend to that of XOS, xylose, and glucose likely due to the ease of arabinose degradation and the structural characteristics of arabinose residues in fibrous materials.

After coupled enzymatic hydrolysis, the yield of XOS increased and showed an overall upward trend, perhaps due to the relatively low concentration of autohydrolysis products and the decline of product inhibition with the resulting increase in the solid-liquid ratio. At the same time, the concentration of other by-products and their inhibitory effects decreased. In addition, with the solid-liquid ratio increase, the total amount of effective hydrogen ions, hydronium ions, and/or acetic acid in autohydrolysis increased, thus degrading the fibrous material structure. These changes are conducive to bonding of xylanase molecules to xylan fragments thus enhancing hydrolysis. The changed trend of the XOS-to-xylose ratio before and after enzymatic hydrolysis was similar, but the degree of the change differed. This effect is due to the joint action of autohydrolysis and enzymatic hydrolysis. At a low solid-liquid ratio, the effect of autohydrolysis on the XOS-to-xylose ratio was greater than enzymatic hydrolysis, whereas at a high solid-liquid ratio the effect of enzymatic hydrolysis was greater. Unlike the effect of previous factors, the XOS proportion after enzymatic hydrolysis increases and then decreases with an increase in the solid-liquid ratio. This result shows that in post-autohydrolysis samples with a high solid-liquid ratio is conducive to the hydrolytic effect of xylanase and the content of XOS increased after enzymatic hydrolysis. However, when the solid-liquid ratio was too high, the content of hemicellulose after autohydrolysis decreased in lower XOS production by enzymatic hydrolysis. After autohydrolysis plus enzymatic hydrolysis treatment, the yield of XOS at a solid-liquid ratio of 1:14 was not significantly higher than that at 1:12, but the ratio of XOS-to-xylose increased by 34%. Consequently, the solid-liquid ratio of 1:14 was selected as the optimum. 

#### 3.2.5. Response Surface Experimental (RSE) Optimized Autohydrolysis Conditions on XOS Yield

Based on the influence of various factors on the yield of XOS in the single factor experimental results, the response surface experimental design was used to analyze the interaction among temperature, time, and solid-liquid ratio to further optimize autohydrolysis conditions for XOS yield (Table 1). Multiple quadratic regression fitting was carried out on the test result data using Design-Expert software. The regression equation is as follows: Y = 20.33 − 3.21A − 1.36B + 0.05C − 5.17AB + 0.35AC + 1.10BC − 6.13A^2^ − 0.6292B^2^ − 2.40C^2^(3)
where Y is the predicted response (yield of XOS), and A, B, and C represent temperature, time, and solid-liquid ratio, respectively.

The statistical significance of Equation (3) was checked by an *F* test analysis, and the analysis of variance (ANOVA) for the response surface quadratic model was generated (Table 2). The *p* value of the obtained model was significant (*p* = 0.0019), while the mismatch term was not significant (*p* > 0.05), indicating that the regression equation has a good fit and describes well the relationship between the various factors and XOS yield. The determination coefficient *R^2^* of the regression equation was 0.9739 and the adjustment coefficient *R^2^*_adj_ was 0.9270, which further shows that the second-order regression equation fits the test well. The correlation between the predicted and actual values of the equation was high. Thus, the equation can be used to determine the optimal conditions for XOS yield.

Based on the *F* value (Table 2), the order of influence of the three factors on the yield of XOS is: temperature > time > solid-liquid ratio, in which the primary terms A and B reach a significant level (*p* < 0.05). Therefore, temperature and time significantly affect XOS yield even when enzymatic hydrolysis is coupled, which is like previous reports on the production of XOS by autohydrolysis only. In addition, A^2^ and C^2^ have significant effects on their surfaces, and there is a highly significant interaction between temperature and time whereas the interaction between other factors is not significant. According to the regression equation, a response surface plot of the interaction of various factors on the yield of XOS was obtained (Figure 5). When the autohydrolysis temperature is low, the yield of XOS after coupled enzymatic hydrolysis increases with time, while at higher treatment temperature the yield of XOS decreases with time (Figure 5a). Moreover, when time is held constant, the yield of XOS first increases and then decreases with increasing temperature. The temperature optimum for XOS yield decreased with increasing time so that higher XOS yield was either at low temperature for a longer time or high temperature for a shorter time. As in previous studies, XOS yield was affected by the interaction of autohydrolysis temperature and time, which jointly determine the degree of hemicellulose dissolution and degradation in *Jiuzao* [33,39]. The content and accessibility of degradable hemicellulose components in the autohydrolysis samples and the by-products that inhibit the enzymatic hydrolysis efficiency will also affect the activity of xylanase, and then affect XOS yield. Therefore, it is necessary to comprehensively consider the autohydrolysis treatment intensity (including temperature and time), not only to ensure that an appropriate amount of hemicellulose is released and the fibrous structure in *Jiuzao* is sufficiently disrupted after autohydrolysis treatment, but also to control the generation of by-products affecting enzymatic hydrolysis efficiency. The response surface describing the relationship of temperature and solid-liquid ratio is convex, indicated that the yield of XOS increases first and then decreases with increases either in temperature or in the solid-liquid ratio (Figure 5b). Thus, a combination of conditions of treatment temperature and solid-liquid ratio is better for high XOS yield. The degradation of the fibrous structure and the release hemicellulose components are reduced at low temperature and solid-liquid ratio in the autohydrolysis process. However, high temperature and solid-liquid ratio would cause excess damage to the fibrous structure and produce many xylanase inhibitors that affect the subsequent enzymatic hydrolysis for XOS production. Therefore, appropriate autohydrolysis temperature and solid-liquid ratio are required to obtain optimal XOS yield. At a constant treatment time, the XOS yield increases then decreases with increasing the solid-liquid ratio (Figure 5c). When the solid-liquid ratio is held constant, the change of XOS yields with treatment time is complex and depends on treatment temperature. When the temperature is low, the XOS yield increased with treatment time. At higher temperature the yield decreased as treatment time increased, probably a result of greater fiber destruction and an increase in the concentration of by-products that inhibit enzymatic hydrolysis.

In conclusion, the best treatment conditions are a balance between XOS produced by autohydrolysis and many factors affecting the subsequent production of XOS via enzymatic hydrolysis. The optimal conditions for XOS yield were identified from the simulation equation generated with the Design-Expert software as follows: temperature of 181.5 °C, time of 20 min, and solid-liquid ratio of 1: 13.6. Under these conditions, the XOS yield was predicted at 21.3%. This prediction was verified from the average value of three repeated experiments where the yield of XOS reaches 21.7%, a value nearly identical to the model’s predicted value. This result shows that the simulation equation has good fit and that the model is reliable. Under the optimized conditions, the yield of XOS after enzymatic hydrolysis was two-fold higher than that without enzymatic hydrolysis, and the ratio of XOS-to-xylose was 232.9%. In addition, although the yield of XOS after response surface optimization was like that after single factor optimization, the treatment intensity Log R_o_ of autohydrolysis was reduced from 3.96 to 3.70. Thus, these autohydrolysis conditions were used for subsequent research.

### 3.3. Optimum Conditions of Enzymatic Hydrolysis

The effect of various physical-chemical parameters on enzymatic hydrolysis was studied to optimize production of XOS. Temperature is a key parameter for enzyme action. Temperature affects the thermal energy of substrate molecules and enzyme activity [44]. In the present study, the effect of various temperatures on XOS production was tested. The production of XOS after 2 h of incubation was significantly higher at 60 or 65 °C than other temperatures (Figure 6a). XynAR was genetically modified for thermal stability and its temperature optimum is 75 °C and its half-life at 80 °C is more than 150 min (unpublished data). These characteristics enable xylanase XynAR to better degrade agricultural waste and generate XOS at relatively high temperature. However, the enzymatic hydrolysis efficiency of XynAR was reduced unexpectedly when the temperature exceeded 65 °C. This yield reduction is likely due the inhibitory effect on xylanase activity of substances produced from autohydrolysis.

The pH is another important factor affecting the enzymatic hydrolysis of xylanase. It affects the activity of xylanase, the ionic state of substances, the adsorption capacity of substrate to xylanase, and the inhibition capacity of inhibitor to xylanase [44]. The hydrolysis reaction was tested at five pH values from 3.0–7.0 at 60 °C for 2 h. The yield of XOS increased initially and later decreased and maximum XOS yield occurred between pH 4.0–5.0 (Figure 6b). The activity of XynAR was stable over the range of 3.0–9.0 and high activity remained at pH range 4.0–6.5 (unpublished data). In addition, the optimum pH of xylanase XynAR for autohydrolysis of *Jiuzao* samples for XOS production differed from its optimum pH. This difference may be related to the effects of substances in the autohydrolysis samples on the xylanase activity of XynAR. The pH can affect these substances and the inhibitory activity of xylanase by these substances was reduced when the pH was relatively acidic.

In enzymatic hydrolysis, the ratio of enzyme to substrate must be considered because of product yield, enzymatic hydrolysis rate, and cost [44]. The effect of varying enzyme loading on the volume basis (5–405 U/mL) was evaluated. Increasing the xylanase concentration from 5 to 15 U/mL produced 23.0% to 30.5% of XOS after 2 h of incubation (Figure 6c). The yield of XOS slightly decreased at higher enzyme concentration probably due to decreased efficiency either because of competition between enzyme molecules for substrates or substrate limitation [32]. Thus, 15.0 U/mL was used for production of XOS in further experiments. 

The production of XOS over time increased as the incubation period increased and a hydrolysis period of 4 h achieved the highest production of XOS (Figure 6d). The yield of XOS was 31.2% after 4 h of incubation and the yield then declined to 30.4% after 8 h of incubation. This decline may have been due to either a decreased level of easily accessible hydrolytic sites in the xylan chain or decreased xylanase activity from end-product inhibition. The increased incubation for production of XOS was not required as it did not enhance production of XOS. Compared to other studies, a hydrolysis period of 4 h in the present study was short due to the use of thermostable xylanase [32,44]. These results suggest that the thermostable xylanase is effective in a SF-LF production system and shows great potential for production of XOS. 

To obtain high XOS yield with more cost-effective conditions, an orthogonal test was conducted using the single factor test results. The enzymolysis condition of A_1_B_3_C_3_D_2_ of 60 °C, pH 5, enzyme loading 25 U/mL, and hydrolysis period 4 h, gave the best yield of XOS at 33.0% (Table 3). According to the R values, in the process of xylanase enzymolysis of SF-LF, the influential of the four factors on XOS yield was in the order: B (pH) > C (enzyme loading) > A (temperature) > D (hydrolysis period) (Table 3). The optimized combination is A_1_B_3_C_2_D_1_ was established from analyzing the K values and a validation test was done in which the yield of XOS was 34.2%, higher than that the highest value listed in Table 3. Therefore, the optimal conditions for XOS production with xylanase XynAR were 60 °C, pH 5, enzyme loading of 15 U/mL, and a hydrolysis period 2 h. The XOS yield from *Jiuzao* was lower than that of previous reports using other substrates [45,46,47]. Aside from the ingredient differences, perhaps production from *Jiuzao* was lower because some hemicellulose structures from rice hulls were decomposed under the combined action of high temperature treatment and microbial activity during the *Baijiu* production process [12,33]. However, *Jiuzao* has certain advantages in the conversion of XOS compared with some other substrates such as brewers’ spent grain, pineapple peel waste, and tobacco stalk (Table 4), and a large amount of *Jiuzao* is produced every year. In addition, *Jiuzao* produced by different *Baijiu* production processes will have different XOS production potential and the same raw materials may produce different XOS due to different sources, pretreatment methods and types of enzymes used. Therefore, different methods may be needed to optimize XOS yields from *Jiuzao* obtained from different *Baijiu* production processes. Primarily due to the hydrolysis characteristics of xylanase XynAR in the present study, X3 and X2 were the main components of the XOS, accounting for 44.5% and 41.0% of the total, respectively. 

### 3.4. FTIR Spectroscopy Analysis

FTIR spectroscopy is a common analytical method used to identify the functional groups and molecular conformations of substances. Thus, the effects of the autohydrolysis protocol and enzymatic hydrolysis were determined by FTIR spectroscopy. For comparison purposes, the spectra of untreated rice hulls were also measured. The different treatment samples and raw material had the similar general spectral profile except some characteristic bands (Figure 7). All samples, including rice hulls, *Jiuzao*, *Jiuzao* after autohydrolysis treatment, and *Jiuzao* after autohydrolysis and enzymatic hydrolysis treatments, had a wide band positioned around 3400 cm^−1^, which indicated the vibrations of the hydrogen bound mainly from cellulose and hemicellulose [71]. Compared with rice hulls, *Jiuzao* had blue shifts and different peak intensity at this frequency band and others, which may be related to the destruction of rice hulls in *Jiuzao* and consequent differences in composition. Aside from main component of rice hull, *Jiuzao* also contains sorghum residue and some fungal mycelium. After autohydrolysis treatment, the absorption intensity near this area decreased significantly due to the disruption of hydrogen bonds, indicating that the cellulose and hemicellulose components in *Jiuzao* were damaged by autohydrolysis. The reduced peak intensity after enzymatic hydrolysis indicated that the hemicellulose was further degraded by xylanase. The band at about 2920 cm^−1^ was related to C-H stretching vibration of methyl and methylene in lignin, cellulose, and hemicellulose [72,73]. Compared with *Jiuzao*, the intensity of the absorption peak decreased and there was a slight blue shift as the treatment procedure progressed (autohydrolysis and enzymatic hydrolysis). This result indicated that the lignin, cellulose, and hemicellulose components in *Jiuzao* were degraded after treatment, especially with the enzymatic hydrolysis treatment. The peak at 1740 cm^−1^ in the rice hull sample is attributed to either the acetyl and uronic ester groups of the hemicelluloses or the ester linkage of carboxylic group of the ferulic and p-coumeric acids of lignin and/or hemicelluloses [72]. After *Baijiu* production was complete, this peak was absent, indicating the cleavage of these bonds. These results show that some hemicellulose in rice hulls was degraded during *Baijiu* production, and this hemicellulose may be the most easily degraded component. This result also explains the formation of ferulic acid and other flavor substances in *Baijiu* and the reasons for the low XOS yield from *Jiuzao*. The C=O stretching bands are attributed to the polysaccharides and hemicellulose or lignin ester groups observed at 1640 cm^−1^ [73]. The change in absorption peak intensity of different treatment samples shows that autohydrolysis and enzymatic hydrolysis are important for the degradation of hemicellulose or lignin in *Jiuzao*. The intensive band at 1514 cm^−1^ is due to aromatic skeletal vibrations in bound lignin [74]. Comparing different treatment samples, this band weakens after enzymatic hydrolysis treatment indicating that some lignin combined with hemicellulose will decompose with the degradation of hemicellulose via enzymatic hydrolysis by xylanase after autohydrolysis. The small bands between −1453 cm^−1^ and −1235 cm^−1^, such as 1453, 1373, and 1234 cm^−1^, represent either C-H and C-O or OH bending vibrations in hemicelluloses [74]. There are slight differences between different samples at these range bands, which indicates that the hemicellulose components have been damaged to varying degrees after different treatments. A similar trend was observed at the 1031 cm^−1^ band for the C-OH stretching and β-glycosidic linkages of the cellulose glucose ring, which indicates that some cellulose components in material are more degradation resistant [73]. In conclusion, the cellulose, hemicellulose, and lignin in the components of *Jiuzao* were damaged after autohydrolysis and enzymatic hydrolysis, which shows the value of these two treatments for treating agricultural waste.

### 3.5. Morphological Structure of Rice Hull by Different Pretreatments

Microscopic differences were observed on the surfaces rice hulls, *Jiuzao*, *Jiuzao* after autohydrolysis treatment, and *Jiuzao* after autohydrolysis treatment and enzymatic hydrolysis treatment (Figure 8). The surface of rice hulls, the main component of *Jiuzao*, were flat and smooth, indicating a solid structure (Figure 8a). Large irregular structures are visible, which may epicuticular wax. The rice hulls that had undergone *Baijiu* production showed a marked change, with a corrugated and slightly broken surface likely induced by high temperature and microbial treatment (Figure 8b). The irregular block structure of the raw rice hull was rougher and partly degraded. Some of the surface structure, which may have been wax and partial easily degraded cellulose, hemicellulose, or lignin components, appeared to have been lost during *Baijiu* production and possibly show an indirect reason for the relatively low XOS yield by autohydrolysis coupled with xylanase treatment. Noticeable differences were present in the autohydrolysis treatment sample. The surface structure was further damaged and the epidermis rough, which would enhance contact between xylanase and the internal structure thus increasing accessibility between enzyme and substrate and the surface area available for enzymatic hydrolysis (Figure 8c). Although the surface of the materials appeared to be eroded, with many dents, it still presents a relatively regular structural outline, which could possibly hinder subsequent enzymatic hydrolysis. The surface anatomy of materials changed significantly after enzymatic hydrolysis with debris, holes, and cracks (Figure 8d). The obvious degradation of structural perhaps shows the effects of enzymatic hydrolysis by xylanase.

## 4. Conclusions

The present study established the potential for producing XOS from *Jiuzao* using autohydrolysis pretreatment and enzymatic hydrolysis with a thermostable xylanase. Treating *Jiuzao* at 181.5 °C for 20 min with solid-liquid ratio of 1:13.6 enabled effective hydrolysis with thermostable xylanase to produce XOS. After optimization of enzymatic hydrolysis conditions, the highest yield of XOS from *Jiuzao* was 34.2% at 60 °C and pH 5 with 15 U/mL of XynAR for 2 h. The process has great promise for practical production of XOS from *Jiuzao*.

## Figures and Tables

**Figure 1 foods-11-02663-f001:**
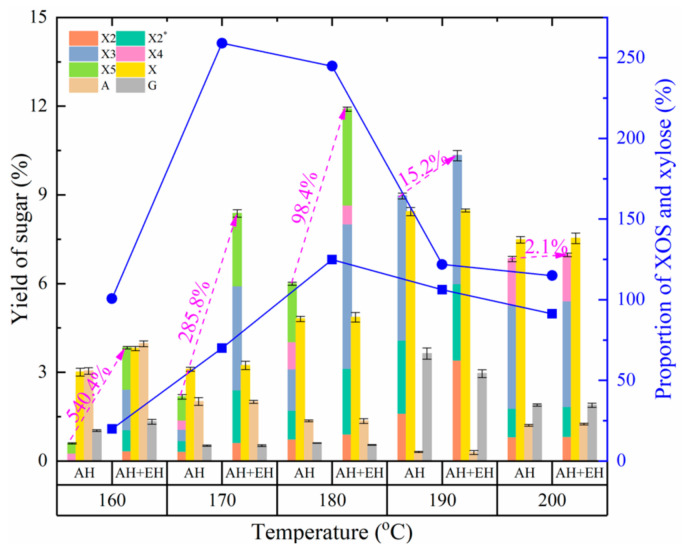
Effects of autohydrolysis pretreatments on sugar yield and the proportion of xylooligosaccharides (XOS) and xylose at different temperatures. AH, autohydrolysis only treatment, no enzymatic hydrolysis treatment; AH + EH, autohydrolysis plus enzymatic hydrolysis treatment. Squares and solid line, the proportion of XOS and xylose for the autohydrolysis-only treatment; circles and solid line, the proportion of XOS and xylose for the autohydrolysis plus enzymatic hydrolysis treatment. The magenta text indicates the percentage increase in the yield of XOS after autohydrolysis plus enzymatic hydrolysis treatment compared with that of the autohydrolysis-only treatment at different temperatures. X2, xylobiose; X2*, an oligosaccharide with a degree of polymerization between X2 and X3; X3, xylotriose; X4, xylotetraose; X5, xylopentaose; X, xylose; A, arabinose; G, glucose.

**Figure 2 foods-11-02663-f002:**
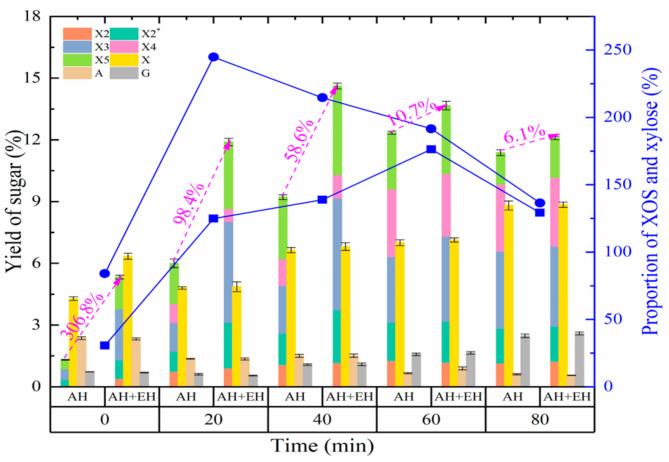
Effects of autohydrolysis pretreatments on sugar yield and the proportion of XOS and xylose over time. AH, autohydrolysis only treatment, no enzymatic hydrolysis treatment; AH + EH, autohydrolysis plus enzymatic hydrolysis treatment. Squares and solid line, the proportion of XOS and xylose for the autohydrolysis-only treatment; circles and solid line, the proportion of XOS and xylose for the autohydrolysis plus enzymatic hydrolysis treatment. The magenta text is the percentage increase in the yield of XOS after autohydrolysis treatment plus enzymatic hydrolysis treatment compared with that of the autohydrolysis-only treatment at different pretreatment times. X2, xylobiose; X2*, an oligosaccharide with a degree of polymerization between X2 and X3; X3, xylotriose; X4, xylotetraose; X5, xylopentaose; X, xylose; A, arabinose; G, glucose.

**Figure 3 foods-11-02663-f003:**
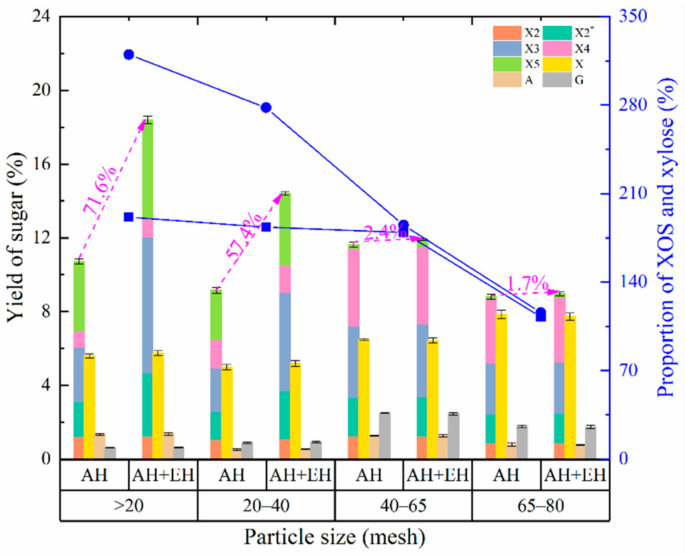
Effects of autohydrolysis pretreatments on sugar yield and the proportion of XOS and xylose at different particle sizes. AH, autohydrolysis-only treatment, no enzymatic hydrolysis treatment; AH + EH, autohydrolysis plus enzymatic hydrolysis treatment. Squares and solid line, the proportion of XOS and xylose for the autohydrolysis only treatment; circles and solid line, the proportion of XOS and xylose for the autohydrolysis plus enzymatic hydrolysis treatment. The magenta text is the percentage increase in the yield of XOS after autohydrolysis plus enzymatic hydrolysis treatment compared with that of the autohydrolysis-only treatment at different particle sizes. X2, xylobiose; X2*, an oligosaccharide with a degree of polymerization between X2 and X3; X3, xylotriose; X4, xylotetraose; X5, xylopentaose; X, xylose; A, arabinose; G, glucose.

**Figure 4 foods-11-02663-f004:**
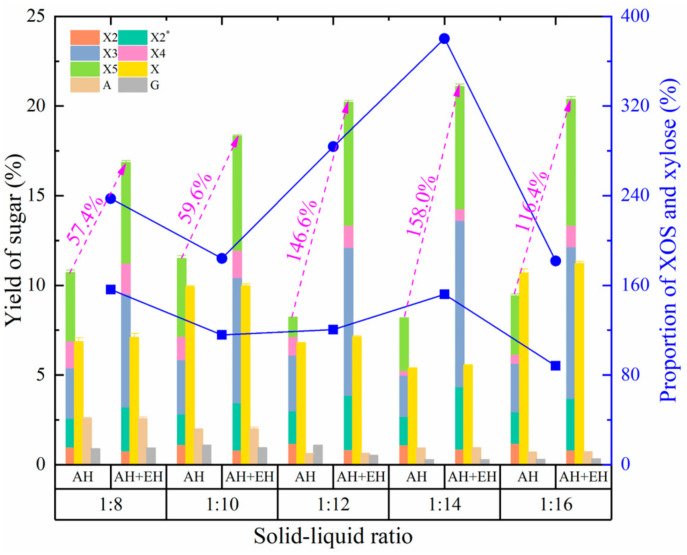
Effects of autohydrolysis pretreatments on sugar yield and the proportion of XOS and xylose at different solid-liquid ratio conditions. AH, autohydrolysis-only treatment, no enzymatic hydrolysis treatment; AH + EH, autohydrolysis plus enzymatic hydrolysis treatment. Squares and solid line, the proportion of XOS and xylose for the autohydrolysis only treatment; circles and solid line, the proportion of XOS and xylose for the autohydrolysis plus enzymatic hydrolysis treatment. The magenta text is the percentage increase in the yield of XOS after autohydrolysis plus enzymatic hydrolysis treatment compared with that of the autohydrolysis-only treatment at different solid-liquid ratios. X2, xylobiose; X2*, an oligosaccharide with a degree of polymerization between X2 and X3; X3, xylotriose; X4, xylotetraose; X5, xylopentaose; X, xylose; A, arabinose; G, glucose.

**Figure 5 foods-11-02663-f005:**
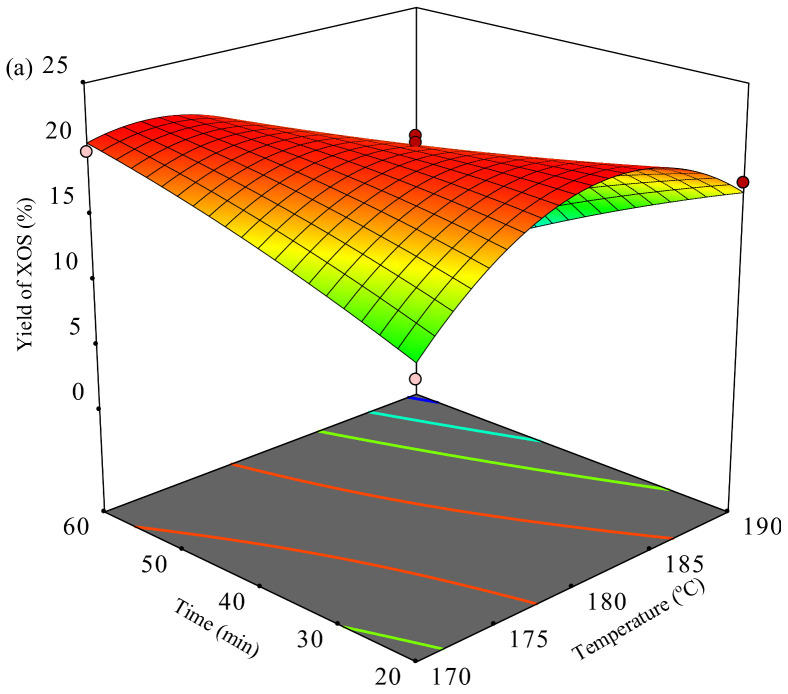
Response surface plot and contour plot for XOS production. (**a**) Effect of interaction of temperature and time on the XOS production when solid-liquid ratio is held at zero level. (**b**) Effect of interaction of temperature and solid-liquid ratio on the XOS production when time is held at zero level. (**c**) Effect of interaction of time and solid-liquid ratio on the XOS production when temperature is held at zero level.

**Figure 6 foods-11-02663-f006:**
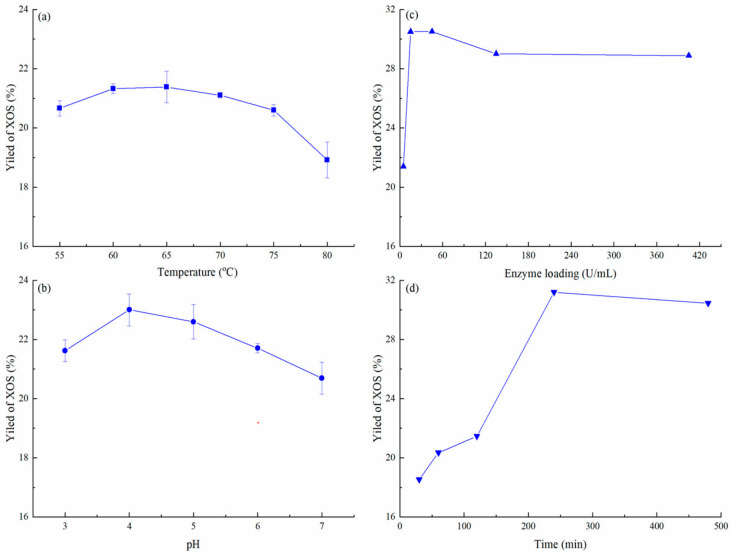
Effect of temperature (**a**), pH (**b**), enzyme loading (**c**) and time (**d**) on XOS production.

**Figure 7 foods-11-02663-f007:**
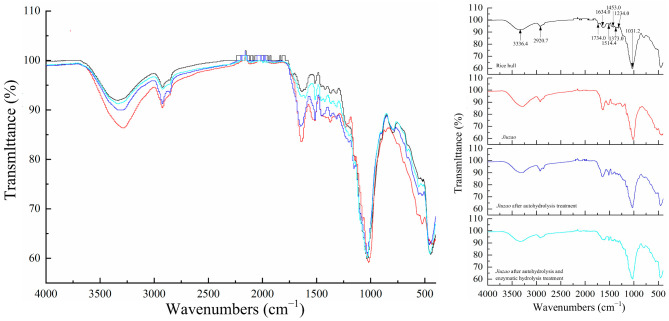
FTIR spectra of rice hulls, *Jiuzao*, *Jiuzao* after autohydrolysis treatment, and *Jiuzao* after autohydrolysis and enzymatic hydrolysis treatments. Right part of chart, combination of FTIR spectra of four samples; left part of chart, FTIR spectra of each of the four samples. Black line, FTIR spectra of rice hulls; red line, FTIR spectra of *Jiuzao*; blue line, FTIR spectra of *Jiuzao* after autohydrolysis treatment; cyan line, FTIR spectra of *Jiuzao* after autohydrolysis and enzymatic hydrolysis treatment.

**Figure 8 foods-11-02663-f008:**
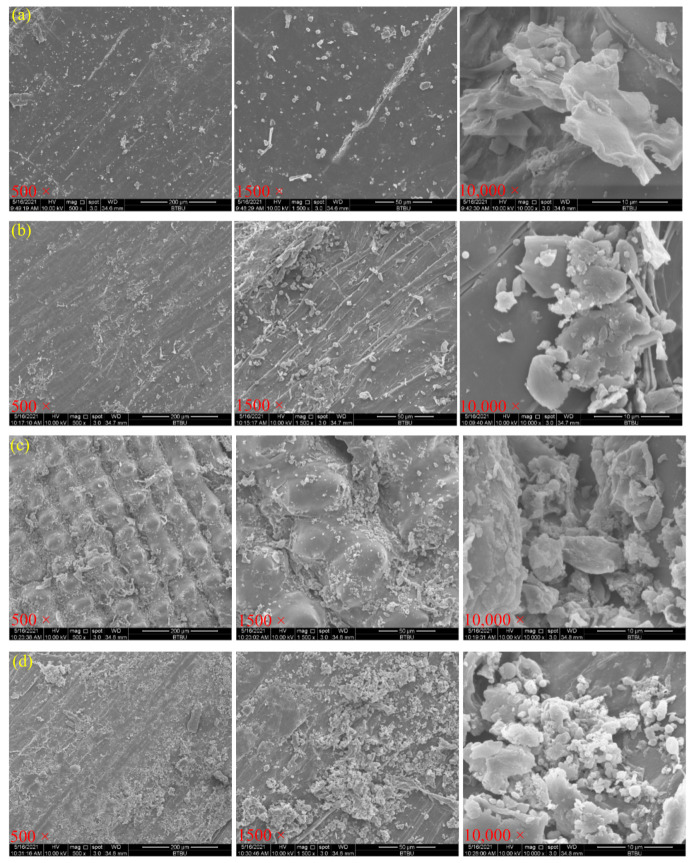
Scanning electron microscope image of rice hulls (**a**), *Jiuzao* (**b**), *Jiuzao* after autohydrolysis treatment (**c**), and *Jiuzao* after autohydrolysis and enzymatic hydrolysis treatment (**d**) at different magnifications.

**Table 1 foods-11-02663-t001:** The Box–Behnken experimental design and the responses of the dependent variables.

Test Number	Temperature (°C)		Time (Min)		Solid-Liquid Ratio		Yield of XOS (%)
	A	Code A	B	Code B	C	Code C	Y
1	190	1	40	0	1:16	1	7.8
2	180	0	20	−1	1:16	1	18.1
3	180	0	60	1	1:12	−1	14.3
4	190	1	40	0	1:12	−1	7.5
5	180	0	40	0	1:14	0	19.3
6	180	0	40	0	1:14	0	21.1
7	180	0	40	0	1:14	0	20.6
8	190	1	20	−1	1:14	0	17.6
9	180	0	20	−1	1:12	−1	19.7
10	180	0	60	1	1:16	1	17.1
11	190	1	60	1	1:14	0	5.0
12	170	−1	20	−1	1:14	0	11.8
13	170	−1	40	0	1:16	1	15.4
14	170	−1	40	0	1:12	−1	16.5
15	170	−1	60	1	1:14	0	19.9

Note: XOS, xylooligosaccharides.

**Table 2 foods-11-02663-t002:** Regression coefficients and their significances for XOS production from the results of the Box-Behnken experimental design.

Source	Coefficient Estimate	Sum of Squares	DF	Mean Square	*F*-Value	*p*-Value	Significant
Model		362.47	9	40.27	20.74	0.0019	**
Intercept	20.33						
A-Temperature	−3.21	82.56	1	82.56	42.52	0.0013	**
B-Time	−1.36	14.85	1	14.85	7.65	0.0396	*
C-Solid-liquid ratio	0.0500	0.02	1	0.02	0.01	0.9231	
AB	−5.17	107.12	1	107.12	55.17	0.0007	**
AC	0.3500	0.49	1	0.49	0.25	0.6368	
BC	1.10	4.84	1	4.84	2.49	0.1752	
A²	−6.13	138.71	1	138.71	71.43	0.0004	**
B²	−0.6292	1.46	1	1.46	0.75	0.4253	
C²	−2.40	21.34	1	21.34	10.99	0.0211	*
Residual		9.71	5	1.94			
Lack of Fit		7.98	3	2.66	3.08	0.2545	not significant
Pure Error		1.73	2	0.86			
Cor Total		372.18	14				
		*R*^2^ = 0.9739	*R*^2^_Adj_ = 0.9270	CV = 9.02%			

Note: “*”, significant at 5% level (*p* < 0.05); “**”, significant at 1% level (*p* < 0.01).

**Table 3 foods-11-02663-t003:** The L9 orthogonal array applied for XOS production.

Test Number	Factors				Yield of XOS (%)
	A (Temperature, °C)	B (pH)	C (Enzyme Loading, U/mL)	D (Time, Min)	
1	1 (60 °C)	1 (3)	1 (5 U/mL)	1 (120 min)	28.6
2	2 (65 °C)	1	2 (15 U/mL)	2 (240 min)	29.0
3	3 (70 °C)	1	3 (25 U/mL)	3 (360 min)	28.0
4	1	2 (4)	2	3	31.3
5	2	2	3	1	31.5
6	3	2	1	2	27.9
7	1	3 (5)	3	2	33.0
8	2	3	1	3	30.4
9	3	3	2	1	32.3
Average K1	31.0	28.5	29.0	30.8	
Average K2	30.3	30.2	30.9	30.0	
Average K3	29.4	31.9	30.9	29.9	
Range (R)	1.6	3.4	1.9	0.9	
Optimal level	1	3	2	1	

Note: XOS, xylooligosaccharides.

**Table 4 foods-11-02663-t004:** Summary of representative XOS productions from different lignocellulosic materials.

Type of Biomass	Hemicellulose (%)	Yield of XOS (%)	References
Rice husk	11.2	69 ^a^	[48]
		33.8 ^b^	[48]
		12.9	[49]
Wheat straw	40	40.3	[50]
	20.1	21	[51]
Rye bran	21	60 ^c^	[52]
		40 ^d^	[52]
Corncob	31.2	75	[53]
	22	57.6	[54]
	33.4	10.7	[29]
Rice straw	24.14	54.3	[55]
	17	50.5	[54]
Mahogany sawdust	24.3	36.8	[31]
Mango sawdust	26.5	25.5	[31]
Sugarcane bagasse	25.7	55.4	[56]
	26.5	53.2	[57]
	20.75	51.1	[58]
	32.7	36.4	[59]
Wheat bran	30	55.9	[60]
	17.6	22.8 ^e^	[61]
		19 ^f^	[61]
		18.6 ^g^	[61]
Hawthorn kernels	28	66.8	[62]
Moso bamboo	17.3	42.7	[63]
Almond shell	20.2	40.6	[64]
Arecanut husk	24.6	35.1	[30]
* Jiuzao *	13.08	34.2	This study
Brewers’ spent grain	16.5	31.5	[65]
Pineapple peel waste	31.8	25.6	[66]
Tobacco stalk	22	11.4	[67]
Natural grass	28	11	[68]
Poplar	15.8	10.7	[69]
Quinoa stalks	10.9	1.26	[70]

Note: ^a^ XOS was produced by *Aspergillus nidulans* XynC A773; ^b^ XOS was produced by *Aspergillus brasiliensis* BLf; ^c^ XOS was produced by RmXyn10A from GH10; ^d^ XOS was produced by Pentopan Mono BG from GH11; ^e^ XOS was produced by Pentopan; ^f^ XOS was produced by RmXyn10A-CM; ^g^ XOS was produced by NpXyn11A.

## Data Availability

All data and materials have been provided in this manuscript.

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
