# Peer review of "Production of Xylooligosaccharides from *Jiuzao* by Autohydrolysis Coupled with Enzymatic Hydrolysis Using a Thermostable Xylanase"

_foods, 2022, doi:10.3390/foods11172663_

Round 1
Reviewer 1 Report
The manuscript focuses on xylooligosaccharide production from an industrial by-product (Jiuzao) which implies the novelty of the work. It’s has been written in a complicated way, especially the interpretations and evaluations of some of the graphs are weak and confusing. There are several badly structured and awkward sentences. Please refer to the specific comments below:
* Line 95: Please describe the procedure for a-amylase treatment.
* Line 130-132: Information and the formula (Eq 1) have been provided for calculation of the severity factor (log Ro). However, nothing has been mentioned about that in “results and discussion section”. If you are not providing any results on that, eliminate this part from “materials and methods” section.
* Line 172: XOS yield calculation (Eq 2)
Please mention specifically in this part that XOS implies oligomers with DP 2-5 for the present study.
* Line 205- 206: “X2*, with a degree of polymerization (DP) between the of X2 and X3”
How did you determine X2* since there is no xylooligosaccharide standard available to be utilized in HPLC analysis for this oligomer? There is no additional method described within the text for identification of this oligomer as well.
* Line 228: “…..with a DP of 2-10 is lower….”
It should be DP of 2-5 as these are the oligomers utilized as the standards in the present study.
* Line 258- 261: “Also, based on the different changes in XOSs and xylose yields after enzymatic hydrolysis, although the ratio of XOSs to xylose after enzymatic hydrolysis still shows a change trend like that with autohydrolysis treatment, the treatment temperature producing the highest XOSs to xylose ratio differed”.
Example of a badly structured interpretation.
* Line 296-301: Another example of a badly structured evaluations.
* Fig 1-4: I suggest removing pink arrows and “increased to….%” as it makes the graphs more complicated. You can keep the percentages at the top.
* Table 2: Which significance level (P < 0.01 or P < 0.05) did you get for generation of the equation 3?
It would have been better to include estimated regression coefficients in ANOVA table so that generation of equation could be observed.
* Line 444-446: “Based on the F value (Table 2), the order of influence of the three factors on the yield of XOSs is: temperature>time>solid-liquid ratio, in which the primary terms A and B reach a significant level (P<0.05)”
According to Table 2; A is significant at level P<0.01. Please correct either the table or the text.
* Line 510-511: “XynAR was genetically modified for thermal stability and its temperature optimum is 75°C and its half-life at 80°C is more than 150 min”
Indicate a reference for this sentence.
* Line 511-513: “These characteristics enable better degradation of agricultural waste by xylanase XynAR to generate XOSs”.
This interpretation does not seem appropriate. Having high optimum temperature for an enzyme does not reflect this. It even means extra energy cost.
* Line 573- 575: “All samples had a wide band positioned around……….”
Identify the samples. It has not been mentioned either here or under “materials and method” section.
* I will suggest not using “XOSs” to indicate the plural form of oligomers. XOS is a widely accepted abbreviation within the researchers working in this field.
Author Response
Responded to all of the comments
Dear Reviewer:
Thank you for your letter and for your comments. The main corrections in the paper and the responds to your comments are as following:
Comments and Responses:
1. The manuscript focuses on xylooligosaccharide production from an industrial by-product (Jiuzao) which implies the novelty of the work. It’s has been written in a complicated way, especially the interpretations and evaluations of some of the graphs are weak and confusing. There are several badly structured and awkward sentences. Please refer to the specific comments below:
Response: Dear reviewer, firstly, thank you very much for your professional opinions. Then, please understand the trouble caused by our unreasonable writing, and thank you for your instruction. Lastly, we have made modifications one by one according to your opinions, and we believe that some improvements can eliminate badly structured and awkward sentences to be easily understood.
2. * Line 95: Please describe the procedure for a-amylase treatment.
Response: Dear reviewer, thank you very much. We have omitted the relevant references cited. The procedure in detail as follows: Jiuzao was smashed and filtered with a 20-40 mesh sieve, then was mixed with 50 mmol/L pH 7.0 phosphate buffer at a proportion of 1:12 and boiled for 30 min; amylase (90 U/mg Jiuzao) from Aspergillus oryzae (Solarbio Science & Technology, Beijing, China) was then added with stirring for 30 min at 90 °C. Finally, the product was dried and smashed for further study. We have reviewed them as follows: “Jiuzao that has been completely hydrolyzed with α-amylase to remove starch according to our previous study [22] was used to assess carbohydrate and lignin content with the method of the National Renewable Energy Laboratory (NREL) [23].”. (Page 3, line 95-97).
3.* Line 130-132: Information and the formula (Eq 1) have been provided for calculation of the severity factor (log Ro). However, nothing has been mentioned about that in “results and discussion section”. If you are not providing any results on that, eliminate this part from “materials and methods” section.
Response: Dear reviewer, so sorry for this. We compared the conditions of autohydrolysis after response surface optimization with those after single factor optimization. Although the yield of XOS was not improved, the treatment intensity of autohydrolysis was reduced, indicating that there was a certain interaction between factors for autohydrolysis. Using severity factor to reflect the reduce of the intensity of autohydrolysis after response surface optimization, and it can achieve the effect of high-yield XOS (Page 19, line 528-531).
4.* Line 172: XOS yield calculation (Eq 2)
Please mention specifically in this part that XOS implies oligomers with DP 2-5 for the present study.
Response: Dear reviewer, thank you very much for this opinion. We have reviewed it as follows: “C(Xi) is the XOS (DP = 2-5)” (Page 4, line 177-180).
5.* Line 205- 206: “X2*, with a degree of polymerization (DP) between the of X2 and X3”
How did you determine X2* since there is no xylooligosaccharide standard available to be utilized in HPLC analysis for this oligomer? There is no additional method described within the text for identification of this oligomer as well.
Response: Dear reviewer, thank you very much. Since there is no corresponding standard for X2*, we calculated it with reference to the standard curve of xylobiose. And, we have reviewed this as follows: “The production of this oligosaccharide was measured according to the standard curve for X2, and it was likely due to the complexity and structure of the fibrous materials and random interruption of the fibrous structure during autohydrolysis.” (Page 5, line 212-215). And, for this substance, our subsequent research has confirmed that it is composed of two xyloses and one arabinose, which is consistent with our speculated structure. At present, we are carrying out yield optimization, probiotic proliferation effect and antioxidant research around this product.
6.* Line 228: “…..with a DP of 2-10 is lower….”
It should be DP of 2-5 as these are the oligomers utilized as the standards in the present study.
Response: Dear reviewer, thank you very much. We have changed as your advice as follows: “When the temperature is low, large soluble polymers and xylose are mostly produced as content of XOS with a DP of 2-5 is lower.” (Page 6, line 234-235).
7.* Line 258- 261: “Also, based on the different changes in XOSs and xylose yields after enzymatic hydrolysis, although the ratio of XOSs to xylose after enzymatic hydrolysis still shows a change trend like that with autohydrolysis treatment, the treatment temperature producing the highest XOSs to xylose ratio differed”.
Example of a badly structured interpretation.
Response: Dear reviewer, thank you very much. We have reviewed them as “Also, based on the different changes in XOS and xylose yields after enzymatic hydrolysis, the ratio of XOS to xylose after enzymatic hydrolysis shows a changed trend unlike that with autohydrolysis treatment, and the treatment temperature producing the highest XOS to xylose ratio differed” (Page 7, line 268-271).
8.* Line 296-301: Another example of a badly structured evaluations.
Response: Dear reviewer, thank you very much. We have reviewed them as “Although the changed behavior of the XOS yield with and without enzymolysis was similar, increasing at first and then decreasing with greater hold time, the yield of XOS after xylanase hydrolysis was higher than that without enzymatic hydrolysis. Moreover, the hold time for the highest yield of XOS was shorter in the autohydrolysis plus enzymatic hydrolysis treatment (at a hold time of 40 min) than that in the autohydrolysis treatment (at a hold time of 60 min). (Page 9, line 309-318)”
9.* Fig 1-4: I suggest removing pink arrows and “increased to….%” as it makes the graphs more complicated. You can keep the percentages at the top.
Response: Dear reviewer, thank you for this good advice. We have reviewed them as your suggestion, but in order to express clearly, we still keep the arrow too and make appropriate adjustments (Page 6, figure 1, line 240-241; page 8, figure 2, line 297-298; page 11, figure 3, line 374-375; page 13, figure 4, line 408-409).
10.* Table 2: Which significance level (P < 0.01 or P < 0.05) did you get for generation of the equation 3?
It would have been better to include estimated regression coefficients in ANOVA table so that generation of equation could be observed.
Response: Dear reviewer, from the results, the equation 3 was significant at the level of 5%. And, we have supplemented the coefficient estimate in Table 2 (Page 15, Table 2).
11.* Line 444-446: “Based on the F value (Table 2), the order of influence of the three factors on the yield of XOSs is: temperature>time>solid-liquid ratio, in which the primary terms A and B reach a significant level (P<0.05)”
According to Table 2; A is significant at level P<0.01. Please correct either the table or the text.
Response: Dear reviewer, thank you. It is right that A is significant at level P<0.01, but B is significant at level P<0.05. So, A and B are significant at level P<0.05. In order to be simple, we have expressed comprehensively them to be significant at level P<0.05.
12.* Line 510-511: “XynAR was genetically modified for thermal stability and its temperature optimum is 75°C and its half-life at 80°C is more than 150 min”
Indicate a reference for this sentence.
Response: Dear reviewer, thank you very much. But we are very sorry, the results in our other manuscript have been submitted many times and is still under review. As your opinion, we have reviewed it as “XynAR was genetically modified for thermal stability and its temperature optimum is 75°C and its half-life at 80°C is more than 150 min, which these results have not yet been published” (Page 20, line 538-540).
13.* Line 511-513: “These characteristics enable better degradation of agricultural waste by xylanase XynAR to generate XOSs”.
This interpretation does not seem appropriate. Having high optimum temperature for an enzyme does not reflect this. It even means extra energy cost.
Response: Dear reviewer, thank you very much. Considering what you said is reasonable, there will be extra energy cost. We have reviewed as “These characteristics enable xylanases XynAR better degradation of agriculture wastes to generate XOS at relatively high temperature” (Page 20, line 540-541). Although high temperature will cause extra energy cost, it also has many advantages, such as reducing the risks of microbial contamination, increasing enzymatic hydrolysis rate, shortening enzymolysis time, leading the economic feasibility.
14.* Line 573- 575: “All samples had a wide band positioned around……….”
Identify the samples. It has not been mentioned either here or under “materials and method” section.
Response: Dear reviewer, thank you very much. This is our mistake. We have reviewed as “All samples, including rice hulls, Jiuzao, Jiuzao after autohydrolysis treatment, and Jiuzao after autohydrolysis and enzymatic hydrolysis treatments, had a wide band positioned around 3400 cm-1, which indicated the vibrations of the hydrogen bound mainly from cellulose and hemicellulose” (Page 24, line 617-620).
15.* I will suggest not using “XOSs” to indicate the plural form of oligomers. XOS is a widely accepted abbreviation within the researchers working in this field.
Response: Dear reviewer, thank you very much. We have reviewed them as your opinion.

Reviewer 2 Report
There is an additional "and" after the author list
Line 35-36: “A mixture of grain and rice hull residues, known as Jiuzao, is left…”
Line 37-39: Please unify the singular and plural forms of the whole sentence.
Line 90: “Jiuzao moisture content was calculated using the weight before and after drying…”
Line 139: “XynAR production and activity were assessed…”
Line 157 and 160: single-factor or single factor? Please unify the expression of the full text
Line 197: production?
Line 82-88: There are many different flavor-types of baijiu. What is the specific type of jiuzao for the sample in this study?
Line 132 Formulas not aligned with text
Figure 1.'s figure caption is mixed with the text
Please reorganize the figure caption in Fig. 5. There are too many "And", which makes it a bit confusing to read
Figure or Fig.? Please unify the expression of all the figure captions
Why use response surface experimental design and orthogonal test to optimize the parameters of jiuzao and enzymatic hydrolysis conditions, respectively. instead of using response surface experimental design for both optimization?
Author Response
Responded to all of the comments
Dear Reviewer:
Thank you for your letter and for your comments. The main corrections in the paper and the responds to your comments are as following:
Comments and Responses:
1. There is an additional "and" after the author list.
Response: Dear reviewer, thank you very much for all your professional opinions. We have changed the “and” to “,” (Page 1, line 5).
2. Line 35-36: “A mixture of grain and rice hull residues, known as Jiuzao, is left…”
Response: Dear reviewer, thank you. We have reviewed it (Page 1, line 37-38).
3. Line 37-39: Please unify the singular and plural forms of the whole sentence.
Response: Dear reviewer, thank you very much. We have reviewed as “Because eight units of Jiuzao are generated during the production of one unit of Baijiu, nearly 100 million tons of solid waste are generated each year from Baijiu production, creating an environmental concern” (Page 1, line 38-40).
4. Line 90: “Jiuzao moisture content was calculated using the weight before and after drying…”
Response: Dear reviewer, thank you very much. We have reviewed as your advices (Page 3, line 92-93).
5. Line 139: “XynAR production and activity were assessed…”
Response: Dear reviewer, thank you very much for your care. We have reviewed them (Page 4, line 143).
6. Line 157 and 160: single-factor or single factor? Please unify the expression of the full text
Response: Dear reviewer, thank you. We have unified as “single factor” in full text.
7. Line 197: production?
Response: Dear reviewer, thank you. It is “production” (Page 5, line 203).
8. Line 82-88: There are many different flavor-types of baijiu. What is the specific type of jiuzao for the sample in this study?
Response: Dear reviewer, thank you. We have used Jiuzao from the Strong-flavor Baijiu production process. We have reviewed as “Jiuzao was provided by Heibei Bancheng Liquor Group, which is an enterprise producing Strong-flavor Baijiu, in 2019, dried at 60 oC for 3 d, and stored in plastic containers at room temperature.” (Page 2, line 84-86).
9. Line 132 Formulas not aligned with text
Response: Dear reviewer, we have modified it (Page 3, line 136).
10. Figure 1.'s figure caption is mixed with the text
Response: Dear reviewer, thank you for this. We have separated them (Page 7, line 251-252).
11. Please reorganize the figure caption in Fig. 5. There are too many "And", which makes it a bit confusing to read
Response: Dear reviewer, thank you for this. We have reviewed as “Fig. 5 Response surface plot and contour plot for XOS production. (a) Effect of interaction of temperature and time on the XOS production when solid-liquid ratio is held at zero level. (b) Effect of interaction of temperature and solid-liquid ratio on the XOS production when time is held at zero level. (c) Effect of interaction of time and solid-liquid ratio on the XOS production when temperature is held at zero level.” (Page 19, line 512-516)
12. Figure or Fig.? Please unify the expression of all the figure captions
Response: Dear reviewer, thank you for this. We have unified them as “Fig.”.
13. Why use response surface experimental design and orthogonal test to optimize the parameters of Jiuzao and enzymatic hydrolysis conditions, respectively. instead of using response surface experimental design for both optimization?
Response: Dear reviewer, thank you. This is due to the inconsistent number of factors for the optimization of the two conditions. Under autohydrolysis conditions, there are mainly three factors, so the optimization effect by response surface method is better. In the optimization of enzymatic hydrolysis conditions, due to a few more factors, and according to some pre-experiments, the enzymatic hydrolysis conditions are basically clear, so the orthogonal experiment method is used.

Reviewer 3 Report
The article foods-1834045 presents results on the optimization of autohydrolysis and enzymatic hydrolysis for an agro-industrial residue from China. The work carried out many experiments, but the optimized condition obtained was not satisfactory, as the XOS yield was very low. The authors attributed many of their results to aspects that were not evaluated in the work. Other questions:
Abstract
Line 24: Change “including” to “evaluating the effects of”;
Introduction
Line 53: I don't understand: “complex structure of lignin and hemicellulose with cellulose”;
Experimental
Lines 102-104: Rephrase this sentence;
Line 122: change “filtering” to “sieving”;
Line 148: change “xylose” to “reducing sugar”;
Line 151: Why did the authors use this mixture? What is the proportion of each?
Line 170: change “M” to “mol/L”;
Results and discussion
Line 193: Figure S1 is unnecessary. The composition must be described in the text;
- Line 195: The percentage of hemicelluloses in this material is very low. This makes its reuse unfeasible for this purpose;
- Line 205: Cannot attribute that X2* is an XOS, as there was no identification by an analytical technique;
- Line 225: High temperatures can convert xylose into furfural. Under all autohydrolysis conditions furfural should have been quantified. This would help to interpret the results;
- Insert error bars in Figures 1, 2, 3, 4, 6c and 6d. Many of the results were discussed without considering deviations. The discussion must be rewritten;
- Line 242-243: Separate text and caption of the figure 1;
- Line 272: Which materials? It is necessary to describe them in the text to facilitate comparison;
- Line 282: This behavior is very strange;
- Line 417: A 1:14 ratio makes the XOS produced much diluted. This must also be discussed, because they needs be purified;
- Lines 471-477: This passage is an example of exaggerating assumptions. This is inappropriate;
Line 511: Enzyme has been genetically modified? Wouldn't it be the microorganism?
- It is necessary to include a table with several works in the literature presenting the hemicellulose content and the XOS yield for various materials, including the data of the present work. This table may indicate that the data obtained were not satisfactory and that Jiuzao has no potential for this application.
Author Response
Responded to all of the comments
Dear Reviewer:
Thank you for your letter and for your comments. The main corrections in the paper and the responds to your comments are as following:
Comments and Responses:
1. The article foods-1834045 presents results on the optimization of autohydrolysis and enzymatic hydrolysis for an agro-industrial residue from China. The work carried out many experiments, but the optimized condition obtained was not satisfactory, as the XOS yield was very low. The authors attributed many of their results to aspects that were not evaluated in the work.
Response: Dear reviewer, first of all, thank you very much for all your professional opinions. Secondly, we apologize for some problems in our manuscript. Our study focuses on the treatment of Jiuzao, a mixture of grain and rice hull residues are left after the solid-state fermentation and solid-state distillation processes are complete for Baijiu production, by autohydrolysis coupled with xylanase, which provides a reference way for the post-treatment of a large number of Jiuzao in China and this study was the first report for XOS production from Jiuzao. Although, our results are not very satisfactory, at the initial stage, we were surprised to find that the yield of XOS obtained by this method was very high, and the conversion rate reached more than 65%, which made us very excited. This once made us recognize the starting point of our research that Jiuzao was treated by microorganisms in Baijiu production process to be easy degraded. Unfortunately, it was only when we processed the data that we found there was an error in the data calculation. We strictly reviewed and checked the data, and did experiments again to be ensure that the previous surprise was caused by the data processing. However, these results were also unexpected. According to our starting point assumption, the rice hull components in Jiuzao have been pretreated by microorganisms and high-temperature treatment during Baijiu produce process, so it is impossible to have such a low conversion rate. For this fact, there may be reasons that we can’t explain for the time being. Like most similar studies, it is also limited to speculation about the results. These explanations can provide a train of thought that may need to be considered for subsequent studies. Our study focus is on the production of XOS from a large amount of Jiuzao through green and environmental protection methods. Although our results are not ideal as some other previous reports, this may be due to the slightly lower hemicellulose content in our samples, and others reasons, this could provide a certain idea for the reuse of other Jiuzao samples. After all, relevant reports show that the hemicellulose content in Jiuzao is 11%-17%, so it has a certain potential to prepare XOS because the annual output of Jiuzao reaches 100 million tons (Wang DD, Wang LY, Wei YX, Cui Q, Sun ZJ (2017) Effects of solid-state fermentation of Nongxiang Baijiu on the recalcitrance of rice husk. Liquor-making Sci Technol (1): 25-29. https://doi.org/10.13746/j.njkj.2016324; Fan ED, Feng MX, Li CY, Wu DG, Chen YF, Xiao DG, Guo XW (2021) Study on improving the quality of distiller's grains feed by steam explosion combined with various microorganisms. J Agri Biotechnol 1(30): 194-206. https://doi.org/DOI:10.3969/j.issn.1674-7968.2022.01.018). Such a huge Jiuzao needs attention, and its components can be divided and used. Due to different production processes, Jiuzao are subject to different degrees of high temperature and microbial treatment during Baijiu production process, which may have different XOS yield and conversion rates. In addition, the different brewing materials of Baijiu will also cause the complexity of the composition of Jiuzao, which causes us to have more uncertainty and speculation in some explanations than the previous single component research. Although our research has unimaginable advantages, there are also some worthy of attention, such as enzymatic hydrolysis can not only improve the yield of XOS, but also reduce autohydrolysis conditions; the effect of particle size on XOS yield. Later, on this topic, we will continue to explore the rules of XOS production from Jiuzao obtained from different flavors Baijiu production and XOS production from Jiuzao obtained from different processes with same flavors Baijiu production. Lastly, as one of your opinions, we summarized some studies on the production of XOS from biomass, and found that the same biomass will have different XOS yields due to different sources, pretreatment methods and enzymes used. In addition, the yield of XOS and conversion rates in our study are in the middle level compared with other previous researches, indicating that our research results have certain practical significance.
2. Abstract
Line 24: Change “including” to “evaluating the effects of”;
Response: Dear reviewer, thank you very much, we have reviewed it as your opinion (Page 1, line 24).
3. Introduction
Line 53: I don't understand: “complex structure of lignin and hemicellulose with cellulose”;
Response: Dear reviewer, thank you very much, we have reviewed it as “However, any such new production process must overcome two problems that limit the accessibility of components to hydrolytic enzymes: the strong crystalline structure of the cellulose and the complex structure crosslinking of lignin and hemicellulose with cellulose in Jiuzao (Page 2, line 51-54).
4. Experimental
Lines 102-104: Rephrase this sentence;
Response: Dear reviewer, thank you very much, we have reviewed it as “The insoluble lignin in residual dry solid biomass was estimated gravimetrically, which the total dry weight (drying at 105 oC to constant weight) subtracted residual ash by heating at 575 oC for 4 h in a muffle furnace” (Page 3, line 105-108).
5. Line 122: change “filtering” to “sieving”;
Response: Dear reviewer, thank you very much, we have reviewed it as your opinion (Page 3, line 126).
6. Line 148: change “xylose” to “reducing sugar”;
Response: Dear reviewer, thank you very much, we have reviewed it as your opinion (Page 4, line 149-152).
7. Line 151: Why did the authors use this mixture? What is the proportion of each?
Response: Dear reviewer, thank you very much. For the mixture, this is because we found in previous studies that if the residue is unevenly distributed, it will bring great experimental error, which may be due to the fact that the residue contains degradable hemicellulose components and other substances that may affect the enzyme activity. However, through our experiments, we found that the content of XOS obtained from the mixed of the solid fraction (SF) and the liquid fraction (LF) was higher than that of the single of the liquid fraction after enzymatic hydrolysis, so we need to carry out the mixed of SF-LF. Every time, it is carried out in proportion to the weight of SF and volume of LF after autohydrolysis. The weight of SF is calculated and weighed according to 15 mL of LF. For example, after autohydrolysis, the LF is 300 mL, and the SF is 45 g. Then take 15 mL of the LF, weigh 2.25 g of SF by the five points sampling method, and mix them evenly and record it as a sample. Through this method, our experimental error is very small.
8. Line 170: change “M” to “mol/L”;
Response: Dear reviewer, thank you very much, we have reviewed it as your opinion (Page 4, line 174).
9. Results and discussion
Line 193: Figure S1 is unnecessary. The composition must be described in the text;
Response: Dear reviewer, thank you very much, we have reviewed it as “In addition, it also contained 10.08% starch, 11.54% ash and 11.18% others.” (Page 5, line 197-198; line 201-202).
10. - Line 195: The percentage of hemicelluloses in this material is very low. This makes its reuse unfeasible for this purpose;
Response: Dear reviewer, thank you very much. As we mentioned above, although the hemicellulose content in Jiuzao is not very high, the output of Jiuzao is large every year, and researchers have focused on the utilization of different components in Jiuzao. Therefore, this research work has certain research significance. In addition, if it can form an industrial chain, we can use Jiuzao after other components used to produce XOS. In this way, the relative content of hemicellulose in used Jiuzao will be increased, which is of great significance to the sustainable development of Baijiu production industry. At the same time, the treatment of Jiuzao is now a major environmental pollution problem that Baijiu producers need to solve. If there is still some income in this process, it is an ideal result for Baijiu producers. Therefore, this is the starting point of our research and the work after communicating with Baijiu producers. From our current results, although the yield of XOS is not very ideal and it makes Jiuzao unfeasible for XOS production, it plays a certain role in environmental protection. Moreover, Jiuzao produced by different Baijiu brewing processes may be different in the production of XOS. In addition, it has been found from previous studies that Jiuzao also have certain potential in the preparation of XOS.
11. - Line 205: Cannot attribute that X2* is an XOS, as there was no identification by an analytical technique;
Response: Dear reviewer, thank you very much. For X2*, we have subsequently optimized the autohydrolysis conditions to produce more, and purified pure product of X2* through activated carbon. It is composed of two units of xylose and one unit of arabinose. Through our subsequent research, it is proved that this substance is XOS, and we have formed new research, mainly focusing on the optimization of autohydrolysis conditions and separation and identification of a XOS, in addition, including its probiotic proliferation and antioxidant function.
12. - Line 225: High temperatures can convert xylose into furfural. Under all autohydrolysis conditions furfural should have been quantified. This would help to interpret the results;
Response: Dear reviewer, we are sorry for this because the experiment plan was focus on the yield of XOS at first, and the purpose of all tests was also around how to improve the content of XOS and reduce the intensity of autohydrolysis through autohydrolysis coupled with xylanase hydrolysis, so the analysis and detection of by-products were neglected. However, in view of the relevant previous results, we only speculate according to previous reports. In the later research on autohydrolysis, ultrasonic coupling xylanase treatment to improve XOS production from Jiuzao, we would detect the by-products produced in this process as your’s opinion.
13. - Insert error bars in Figures 1, 2, 3, 4, 6c and 6d. Many of the results were discussed without considering deviations. The discussion must be rewritten;
Response: Dear reviewer, thank you very much for this. We have insert error bars for figures 1, 2, 3 and 4 (Page 6, figure 1, line 240-241; page 8, figure 2, line 297-298; page 11, figure 3, line 374-375; page 13, figure 4, line 408-409). In view of our previous pretreatment of autohydrolysis samples, the error of our experimental results is a bit and there are a few targets, which are the reasons why we did not add error at that time for these figures. And, we have rewritten some of discussion. In addition, the figures 6c and 6d have error bars and it was a bit, so that it cannot be observed. This is also the confirmation of our previous experimental method, which can reduce the experimental error.
14. - Line 242-243: Separate text and caption of the figure 1;
Response: Dear reviewer, thank you very much for this. We have separated them (Page 7, line 251-252).
15. - Line 272: Which materials? It is necessary to describe them in the text to facilitate comparison;
Response: Dear reviewer, thank you very much. These materials are explained in the following examples (Page 7, line 282-285).
16. - Line 282: This behavior is very strange;
Response: Dear reviewer, thank you very much. It may be that what we described was inappropriate. In fact, it was slightly reduced first and then increased. Therefore, we have modified as “The glucose yield change showed an initial slight decrease afollowed by an increase.” (Page 7-8, line 291-293).
17. - Line 417: A 1:14 ratio makes the XOS produced much diluted. This must also be discussed, because they needs be purified;
Response: Dear reviewer, thank you very much. It is indeed much diluted in 1:14 ratio compared with 1:12 ratio, but from the perspective of the ratio of XOS to xylose, it was relatively the highest in 1:14 ratio, which was conducive to purify XOS.
18. - Lines 471-477: This passage is an example of exaggerating assumptions. This is inappropriate;
Response: Dear reviewer, thank you very much. We have reviewed them as “The degradation of the fibrous structure and the release hemicellulose components are reduced at low temperature and solid-liquid ratio in the autohydrolysis process. However, under high temperature and solid-liquid ratio would cause excess damage to the fibrous structure and producing many xylanase inhibitors that affect the subsequent enzymatic hydrolysis for XOS production.” (Page 16, line 490-497)
19. Line 511: Enzyme has been genetically modified? Wouldn't it be the microorganism?
Response: Dear reviewer, yes, it is modified, and we using the enzyme to hydrolysis, not microorganism.
20. - It is necessary to include a table with several works in the literature presenting the hemicellulose content and the XOS yield for various materials, including the data of the present work. This table may indicate that the data obtained were not satisfactory and that Jiuzao has no potential for this application.
Response: Dear reviewer, thank you very much for this. We have added the table with several works for various materials according to your comments. Our research is indeed a little low in the yield of XOS compared with some substrates, but it still has some advantage to produce XOS, and as we mentioned earlier, one of our purposes is to provide some ideas for the resource utilization of waste generated in the brewing process of Baijiu. In addition, we also hope to obtain better results in Jiuzao from other Baijiu sources, and provide a reference for whether Jiuzao obtained from different flavor types or different processes Baijiu production in China is suitable for the production of XOS. In addition, as Table 4 mentioned, Jiuzao has certain potential in the production of XOS, but further research is still needed (Page 22, line 594-601; page 22, Table 4).

Round 2
Reviewer 3 Report
The article foods-1834045 presented improvements in its content. However, most of the answers to the questions I asked should be answered/included in the text. For example, the answers to items 7, 10, 11, 17, and 20. In addition, the text still has several language errors. The text must be reviewed by a native English speaker.
In its present form the article cannot be published in this renowned journal.
Author Response
Dear Reviewer:
Thank you for your letter and for your comments. The main corrections in the paper and the responds to your comments are as following:
Comments and Responses:
- The article foods-1834045 presented improvements in its content. However, most of the answers to the questions I asked should be answered/included in the text. For example, the answers to items 7, 10, 11, 17, and 20. In addition, the text still has several language errors. The text must be reviewed by a native English speaker.
In its present form the article cannot be published in this renowned journal.
Response: Dear reviewer, firstly, thank you very much for your approval of our last revision, which is due to the opinions of editors and yours. Secondly, we are very sorry that there are still deficiencies in the last revision. We have responded to your suggestions one by one and reflected the relevant replies in the manuscript. And again, our manuscript has been reviewed by a native English speaker. We hope that this revision would meet the requirements of our journal.
- (last question 7) Line 151: Why did the authors use this mixture? What is the proportion of each?
Response: Dear reviewer, we have reviewed as follows in text: Preliminary experiments showed that a mixture of the SF and LF yielded more XOS after enzymatic treatment than did the SF alone. Therefore, to obtain maximum XOS yield and improve experimental repeatability both the SF and LF were used in subsequent work. The enzymatic hydrolysis with XynAR was performed as described previously [26]. In details, enzymatic hydrolysis was performed on 15 mL samples. The amount of SF required for each 15 mL of LF was calculated to match the proportion in the original autohydrolysis product. After mixing each portion of SF with 15 mL of LF (SF-LF), the 15 mL mixtures were used for enzymatic hydrolysis reactions at 60 oC (Page 4, line 160-168).
- (last question 10) - Line 195: The percentage of hemicelluloses in this material is very low. This makes its reuse unfeasible for this purpose;
Response: Dear reviewer, we have reviewed as follows in text: Although, the hemicellulose content of Jiuzao at the present study was lower than that reported for most other biomass sources, the yearly output of Jiuzao is large and it is the main pollutant generated during Baijiu production [29-31]. Using various Jizuao com-ponents to produce value-added products will help mitigate this environmental prob-lem. Hemicellulose would likely be the first component to be used commercially, with the gradual utilization of other components established through a step-by-step ap-proach. The hemicellulose components in Jiuzao from various Baijiu production facilities could differ and thus the Jiuzao from some plants might be better for XOS production [27, 28] (Page 5, line 215-223).
- (last question 11) - Line 205: Cannot attribute that X2* is an XOS, as there was no identification by an analytical technique;
Response: Dear reviewer, because we have tried to do other manuscripts for the optimization of autohydrolysis conditions, separation, identification, function of X2*, so we just reviewed as follows: X2* was obtained with high performance liquid chromatography according to the XOS analytical method. After treatment with 4% (w/w) sulfuric acid at 121 oC for 1 h, the monosaccharide composition of X2* determined by HPLC to be xylose and arabinose, which confirmed that it was an XOS containing arabinose. The production of this oligosaccharide was measured according to the standard curve for X2, and it was likely due to the complexity and structure of the fibrous materials and random interruption of the fibrous structure during autohydrolysis. Therefore, because of the distribution of arabinose residues in Jiuzao and their sensitivity to high temperature, XOS containing arabinose can only be produced under appropriate autohydrolysis conditions. In futures studies the preparation conditions, purification, the effects of X2* on probiotics and antioxidant function will be studied. A similar trend for xylose yield with XOS yield was reported previously [32] (Page 5-6, line 235-245).
- (last question 17) - Line 417: A 1:14 ratio makes the XOS produced much diluted. This must also be discussed, because they needs be purified;
Response: Dear reviewer, we have reviewed as follows in text: However, when the solid-liquid ratio was too high the content of hemicellulose after autohydrolysis decreased in lower XOS production by enzymatic hydrolysis. After autohydrolysis plus enzymatic hydrolysis treatment, the yield of XOS at a solid-liquid ratio of 1:14 was not significantly higher than that at 1:12, but the ratio of XOS to xylose in-creased by 34%. Consequently, the solid-liquid ratio of 1:14 was selected as the optimum (Page 11, line 461-466).
- (last question 20). - It is necessary to include a table with several works in the literature presenting the hemicellulose content and the XOS yield for various materials, including the data of the present work. This table may indicate that the data obtained were not satisfactory and that Jiuzao has no potential for this application.
Response: Dear reviewer, we have added a table including works as our work and the table included the hemicellulose content and the XOS yield. Then, some discussions have be supplied as follows: The XOS yield from Jiuzao was lower than that of previous reports using other substrates [45-47]. Aside from the ingredient differences, perhaps production from Jiuzao was lower because some hemicellulose structures from rice hulls were decomposed under the combined action of high temperature treatment and microbial activity during the Baijiu production process [12, 33]. However, Jiuzao has certain advantages in the conversion of XOS compared with some other substrates such as brewers' spent grain, pineapple peel waste, and tobacco stalk (Table 4), and a large amount of Jiuzao is produced every year. In addition, Jiuzao produced by different Baijiu production processes will have different XOS production potential and the same raw materials may produce different XOS due to different sources, pretreatment methods and types of enzymes used. Therefore, different methods may be needed to optimize XOS yields from Jiuzao obtained from different Baijiu production processes. Primarily due to the hydrolysis characteristics of xylanase XynAR in the present study, X3 and X2 were the main components of the XOS, accounting for 44.5% and 41.0% of the total, respectively. (Page 17-18, line 613-628).
Dear reviewer, thank you very much again for your attention and consideration again. If there are still deficiencies in our manuscript, please do not hesitate to comment.